# 🔥 FLAME-*in*-NeRF 🔥: Neural control of Radiance Fields for Free View Face Animation

## Abstract

This paper presents a neural rendering method for controllable portrait video synthesis. Recent advances in volumetric neural rendering, such as neural radiance fields (NeRF), have enabled the photorealistic novel view synthesis of static scenes with impressive results. However, modeling dynamic and controllable objects as part of a scene with such scene representations is still challenging. In this work, we design a system that enables 1) novel view synthesis for portrait video, of both the human subject and the scene they are in and 2) explicit control of the facial expressions through a low-dimensional expression representation. We represent the distribution of human facial expressions using the expression parameters of a 3D Morphable Model (3DMMs) and condition the NeRF volumetric function on them. Furthermore, we impose a spatial prior, brought by 3DMM fitting, to guide the network to learn disentangled control for static scene appearance and dynamic facial actions. We show the effectiveness of our method on free view synthesis of portrait videos with expression controls. To train a scene, our method only requires a short video of a subject captured by a mobile device.

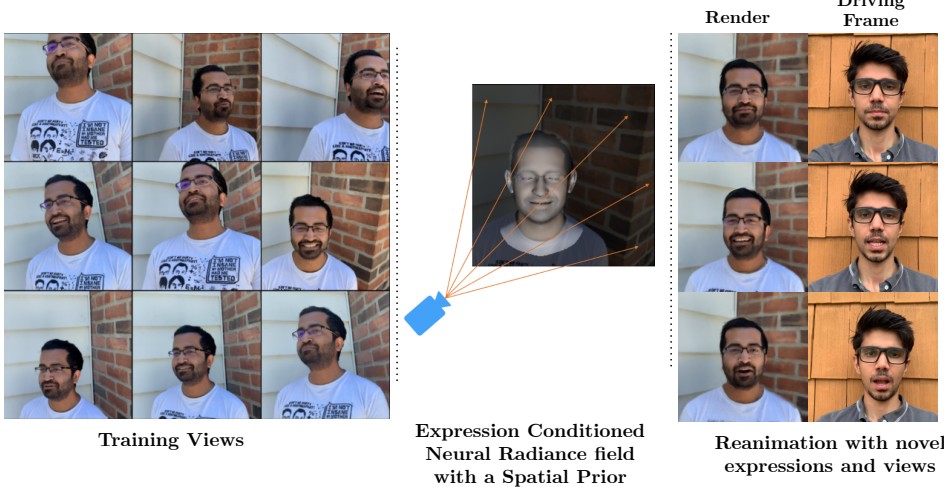

Figure 1: **FLAME-*in*-NeRF.** Our method, FLAME-*in*-NeRF, models portrait videos (left) using an expression conditioned neural radiance field with a spatial prior (middle). Once trained, FLAME-*in*-NeRF can reanimate the subject and the scene present in the portrait video with arbitrary facial expressions and novel views.

## 1 Introduction

There is still no fully controllable human head model in natural scenes with arbitrary view synthesis, despite recent increased research. Such a model, in principle, allows for arbitrary control of human head pose, facial expression, identity and viewing direction. Earliest attempts towards a fully controllable human head model were in the form of 3D Morphable Models (3DMMs) (Blanz et al., 1999). 3DMMs use a PCA-based space to independently control face shape, facial expressions and

appearance and can be rendered in any view using standard graphics-based rendering techniques such as rasterization or ray-tracing. However, 3DMMs (Blanz et al., 1999) lack the ability to capture fine details of the human head such as hair, skin details and accessories such as glasses. Additionally, only the 3D face can be viewed in novel directions, the scene itself cannot, as the mesh only models the human head and nothing else. In contrast, recent methods for novel view synthesis of static and dynamic scenes (Mildenhall et al., 2020; Park et al., 2021; Gafni et al., 2020; Chibane et al., 2021; Li et al., 2021b; Pumarola et al., 2021; Zhang et al., 2020b; Gao et al., 2020; Liu et al., 2020; Zhang et al., 2020a; Bemana et al., 2020; Martin-Brualla et al., 2020; Xian et al., 2021) are able to generate high quality novel views of a given captured scene but lack any control of the objects contained within the scene, including that of the human face and its various attributes.

In this paper we introduce FLAME-*in*-NeRF, a method that is capable of arbitrary facial expression control and novel view synthesis. We represent the whole scene as a neural radiance field in a manner similar to (Mildenhall et al., 2020; Park et al., 2021; Gafni et al., 2020) and lend it explicit expression controls using expression parameters derived from a morphable model (Li et al., 2017). A simple way to utilize the expression parameters within a NeRF (Mildenhall et al., 2020), is to concatenate them to the color-MLP as done in Gafni et al. (2020). However, as shown in Fig 2, such a naive-concatenation entangles the appearance of the scene to the expression parameters causing significant artefacts during reanimation. Standard volumetric representations, such as those used in vanilla NeRFs (Mildenhall et al., 2020), do not distinguish between various objects of the 3D scene since they lack any object semantics for scene they're representing. Consequently, naively concatenating controlling parameters, in this case expression parameters, leads to the appearance being entangled with the controlling parameters. In order to avoid this entanglement we, once again, utilize the 3DMM to impose a spatial prior on the 3D scene. Such a prior ensures explicit disentanglement between appearance and expression parameters in parts of the 3D scene where we know the human head is not present, ensuring that the appearance of scene points that do not project on the human face are unaffected by changes in expression. Our model is trained on videos captured using a mobile phone, either by oneself or by someone else. Once trained, FLAME-*in*-NeRF allows for explicit control of both facial expression and viewing direction while capturing rich details of the scene along with fine details of the human head such as the hair, beard, teeth and accessories such as glasses. Videos reanimated using our method maintain high fidelity to both the driving video in terms of facial expression manifestation and the original captured scene and human head.

In summary, our contributions in this paper are as follows: 1) We propose a first-of-a-kind neural radiance field capable of explicit dynamic control on objects, such as the human face, within a static scene. 2) We experimentally demonstrate expression-appearance entanglement when reanimating portrait videos using standard neural radiance fields. 3) We introduce a spatial ray sampling prior that ensures explicit disentanglement between facial expressions and appearance and significantly improves quality of reanimation. 4) We develop a system capable of simultaneous control of facial expressions and viewing direction trained on short videos captured from a mobile phone.

## 2    RELATED WORK

FLAME-*in*-NeRF is a method for arbitrary facial expression control and novel view synthesis of scenes captured in portrait videos. It is closely related to recent work on neural rendering and novel view synthesis, 3D face modeling, and controllable face generation.

**Neural Scene Representations and Novel View Synthesis.**   FLAME-*in*-NeRF is related to recent advances in neural rendering and novel view synthesis (Mildenhall et al., 2020; Park et al., 2021; Gafni et al., 2020; Yariv et al., 2020; Chibane et al., 2021; Lassner & Zollhöfer, 2021; Lombardi et al., 2021; Li et al., 2021b; Pumarola et al., 2021; Riegler & Koltun, 2021; Zhang et al., 2020b; Oechsle et al., 2021; Gao et al., 2020; Sitzmann et al., 2019; Liu et al., 2020; Zhang et al., 2020a; Bemana et al., 2020; Martin-Brualla et al., 2020; Xian et al., 2021; Wizadwongsa et al., 2021). Notably, Neural Radiance Fields (NeRF) uses a Multi-Layer Perceptron (MLP), $F$, to learn a volumetric representation of a scene. Given a 3D point and the direction from which the point is being viewed, $F$ predicts its color and volume density. For any given camera pose, $F$ is first evaluated densely enough throughout the scene using hierarchical volume sampling (Mildenhall et al., 2020), then volume rendering is used to render the final image. $F$ is trained by minimizing the error between the predicted color of a pixel and its ground truth value. While NeRF is able to generate high quality and

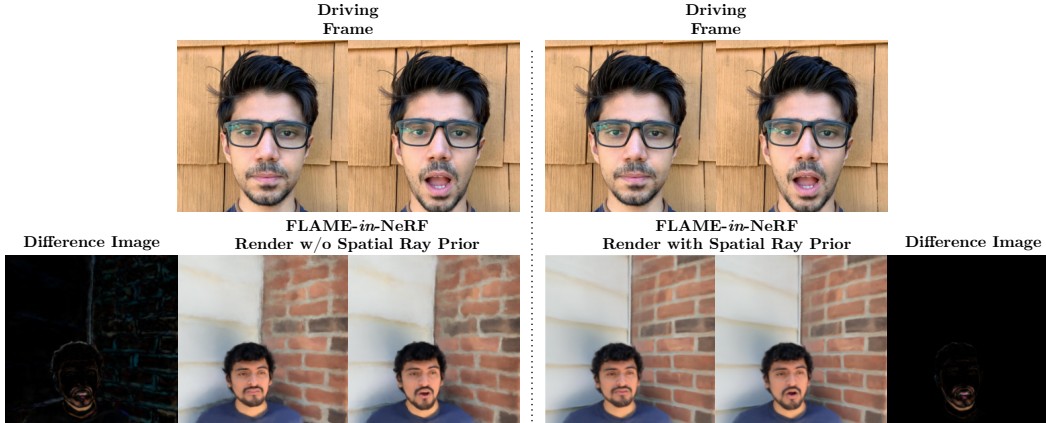

Figure 2: **Expression-Appearance Entanglement: Why use the Spatial Ray Prior?**: Here we demonstrate the necessity of using a Spatial Ray Prior for the reanimation of portrait videos with arbitrary facial expressions and view control. On the left we have a model that does not use a Spatial Ray Prior and on the right, a model that does. As can be seen, the model without the prior generates results of lower quality (e.g. on the lines of the brick wall) than the model with it. Further, the difference images show that, *despite keeping the viewing direction constant*, the model without the spatial prior changes the background appearance with changing expression. In contrast, the model with the spatial ray prior does not do so as the prior explicitly disentangles the expression and the appearance in regions of the 3D scene that do not project to the face. *(Please watch the accompanying video in Supplementary)*.

photo-realistic images for novel view synthesis, it is only designed for a static scene and is unable to represent scene dynamics. Specifically designed for dynamic portrait video synthesis, our approach not only models the dynamics of human faces, but also allow specific controls on the facial animation.

**Dynamic Neural Scene Representations.**  Methods such as (Li et al., 2021b;a; Pumarola et al., 2021; Xian et al., 2021) extend NeRF to dynamic scenes by providing as input a time component and along with it imposing temporal constraints using scene flow (Li et al., 2021b; Xian et al., 2021) or by using a canonical frame (Pumarola et al., 2021). Similarly, Nerfies (Park et al., 2021) too work with dynamic scenes by mapping to a canonical frame, however it assumes that the movement is small. FLAME-*in*-NeRF, like (Park et al., 2021), models the portrait video by mapping to a canonical frame and assumes that the head motion in the video is small.

**Controllable Face Generation.**  The advent of adversarial training (Goodfellow et al., 2014), cycle-consistency losses (Zhu et al., 2017) and powerful convolutional architectures (Isola et al., 2017) have made it possible to perform high quality facial expression editing just using 2D images (Shu et al., 2018; 2017; Athar et al., 2020a; Pumarola et al., 2020; Choi et al., 2018; 2020). However, since these methods are restricted to images and are often trained on frontal datasets, their quality degrades as the pose of the input image changes. Methods such as (Kim et al., 2018; Doukas et al., 2021; Koujan et al., 2020; Athar et al., 2020b) that use a 3DMM to reanimate faces. While being able to do so with great detail, they are unable to perform novel view synthesis as they do not model the geometry of the whole scene. In (Gafni et al., 2020), the authors use neural radiance fields to provide full control on the head. However, they do not model the background and it is assumed to be static. In contrast, FLAME-*in*-NeRF provides full control over the facial expressions of the person captured in the portrait video and has to ability to synthesize novel views. However, FLAME-*in*-NeRF does not provide control over the head-pose and we leave that to future work.

## 3  FLAME-*in*-NeRF

In this section, we describe our method, FLAME-*in*-NeRF, that enables novel view synthesis of dynamic portrait scene and arbitrary control of facial expressions. We model a dynamic portrait scene using a Neural Radiance Field with per-point deformation (Park et al., 2021) to allow for slight movement of the head. The deformation mechanism introduced in Park et al. (2021) deforms the

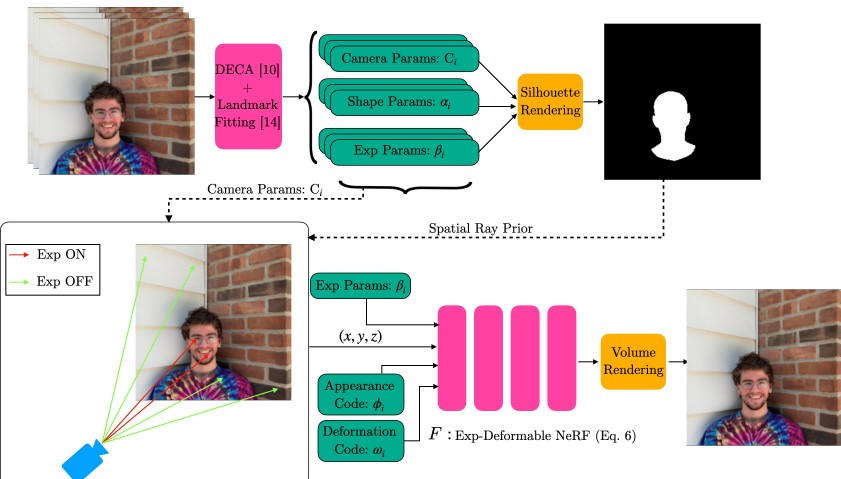

Figure 3: **Overview of training FLAME-*in*-NeRF.** First, we use DECA (Feng et al., 2021) and landmark fitting (Guo et al., 2020) to extract per-frame camera, shape, and expression parameters. Next, these parameters are used to render a silhouette of the FLAME model geometry. This silhouette is used to provide a spatial prior on ray sampling where only points that lie on rays that intersect the silhouette are affected by the expression parameters. Finally, given the $i$-th frame, we shoot rays, we sample points along them and input these points to the Deformable NeRF, $F$, along with the $i$-th frame's expression parameters, deformation code and appearance code to render the final image.

rays of each frame to a canonical frame in order to ensure the rays that intersect are photometrically consistent. Facial expression dynamics are controlled by per-frame FLAME expression parameters (Li et al., 2017) derived using (Feng et al., 2021) followed by standard landmarks fitting. In order to ensure disentanglement between the view parameters and the expression parameters, we adopt spatial prior on ray sampling during training. Specifically, we use a silhouette rendering of the fitted FLAME face model and exclude the expression parameters for all points on rays that do not intersect the silhouette.

### 3.1 DEFORMABLE NEURAL RADIANCE FIELDS

A neural radiance field (NeRF) is a continuous function of, $F : (\boldsymbol{\gamma}(\mathbf{x}), \boldsymbol{\gamma}(\mathbf{d})) \rightarrow (\mathbf{c}(\mathbf{x}, \mathbf{d}), \sigma(\mathbf{x}))$, that, given a 3D point of a scene $\mathbf{x}$ and the viewing direction $\mathbf{d}$ (i.e the direction of the ray it is on) gives the color $\mathbf{c} = (r, g, b)$ and the density $\sigma$. Here, $F$ is a multi-layer perceptron (MLP) and $\boldsymbol{\gamma} : \mathbb{R}^3 \rightarrow \mathbb{R}^{3+6m}$ is the positional encoding (Mildenhall et al., 2020) defined as $\boldsymbol{\gamma}(\mathbf{x}) = (\mathbf{x}, ..., \sin(2^k \mathbf{x}), \cos(2^k \mathbf{x}), ...)$ where $m$ is the total number of frequency bands and $k \in \{0, ..., m-1\}$. The expected color of a camera ray $\mathbf{r}(t) = \mathbf{o} + t\mathbf{d}$, where $\mathbf{o}$ is the camera center and $\mathbf{d}$ is the direction of the ray, is given by the standard volumetric rendering equation

$$C(\mathbf{r}) = \int_{t_n}^{t_f} T(t)\sigma(\mathbf{r}(t))\mathbf{c}(\mathbf{r}(t), \mathbf{d})dt, \text{ where } T(t) = \exp\left(-\int_{t_n}^{t} \sigma(\mathbf{r}(s))ds\right) \quad (1)$$

where, $T(t)$ is the accumulated transmittance along the ray from $t_n$ to $t$. In practice, the integral in Eq. (1) is estimated using hierarchical volume sampling, we refer the reader to (Mildenhall et al., 2020) for details. Given multiple images of a scene, with their associated camera intrinsics and extrinsics, rays are shot through each pixel of each image of the scene using mini-batches. The color of each ray is accumulated via volume rendering using Eq. (1) and the error w.r.t the ground truth pixel color is minimized as follows:

$$\min_{\theta} \sum_{p} ||C_p(\theta) - C_p^{GT}|| \quad (2)$$

where, $p$ is an indexing variable over all the pixels in the mini-batch, $\theta$ are the parameters of $F$ and $C_p^{GT}$ is the ground truth pixel color.

NeRF, as defined above, is naturally designed for static scenes where the assumption is that two rays that intersect would have the same color. However, as observed in (Park et al., 2021), humans rarely

remain perfectly static during a video capture process, more so when they are speaking or performing facial expressions, thus vanilla NeRF training fails to model such dynamic scenes. In order to take into account for subtle movement of subjects in the capturing process, Park et al. (2021) proposed the Deformable NeRF architecture. In Park et al. (2021), 3D points, $\mathbf{x}$'s, captured in the $i$-th frame of the video are deformed to a canonical space via a deformation function $D_i : \mathbf{x} \to \mathbf{x}'$. Here, $D_i$ is defined as $D(\mathbf{x}, \omega_i) = \mathbf{x}'$ where $\omega_i$ is a per-frame latent deformation code. In practice, $D(\mathbf{x}, \omega_i)$ is modeled using an MLP and its coordinate input is also positionally encoded, we choose to omit it for brevity. In addition to a deformation code, $\omega_i$, Park et al. (2021) also uses a per-frame appearance code, $\phi_i$, thus the final radiance field for the $i$-th frame is as follows:

$$F : (\boldsymbol{\gamma}(D(\mathbf{x}, \omega_i)), \boldsymbol{\gamma}(\mathbf{d}), \phi_i) \to (\mathbf{c}(\mathbf{x}, \mathbf{d}), \sigma(\mathbf{x})) \tag{3}$$

In addition to the parameters of $F$ each $\omega_i$ and $\phi_i$ are also optimized through stochastic gradient descent. In practice, $D(\mathbf{x}, \omega_i)$ is modeled as a dense SE(3) field and we use coarse-to-fine regularization on the co-ordinate input, we refer the reader to Park et al. (2021) for details. While the aforementioned methods are able generate novel views (Mildenhall et al., 2020; Park et al., 2021) and handle small movement of objects in the scene (Park et al., 2021), they are still unable to control them.

## 3.2 EXPRESSION CONTROL IN DEFORMABLE NEURAL RADIANCE FIELDS

FLAME-*in*-NeRF models changes of subject's facial expression as changes in the color of 3D scene points. To this end, we condition the learning of deformable NeRF on a set of expression parameters provided by FLAME (Li et al., 2017). The expression conditioned deformable NeRF is defined as follows:

$$F : (\boldsymbol{\gamma}(D(\mathbf{x}, \omega_i)), \boldsymbol{\gamma}(\mathbf{d}), \phi_i, \beta_i) \to (\mathbf{c}(\mathbf{x}, \mathbf{d}), \sigma(\mathbf{x})) \tag{4}$$

where, $\beta_i$ is the expression parameter of the $i$-th frame, $D$ is the deformation function, $\omega_i$ and $\phi_i$ are the deformation code and appearance code of the $i$-th frame respectively. The expected colors of each pixel are then calculated using Eq. (1).

## 3.3 SPATIAL PRIOR FOR RAY SAMPLING

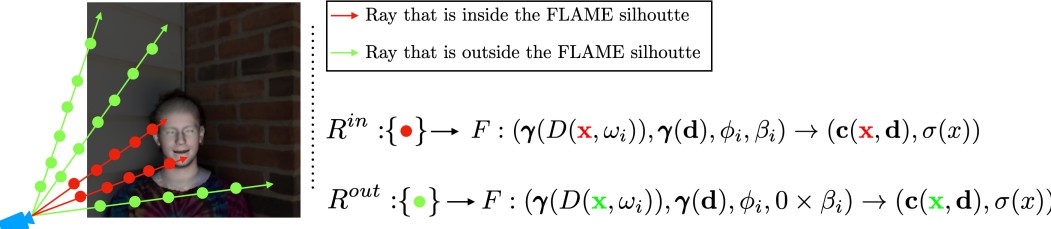

Figure 4: **FLAME induced Prior.** FLAME-*in*-NeRF uses a silhouette rendering of the FLAME model geometry (an overlay is shown above) to provide a spatial prior on rays shot through the 3D scene. All points that lie on rays that intersect the silhouette, shown in red, are affected by the expression parameters. Other points, shown in green, have their expression parameters set to zero and are therefore unaffected by changes in expression.

When the radiance field is modeled as described in Eq. (4), there is nothing that prevents the appearance of a point $\mathbf{x}$, that does *not* project on the face, to become dependent on the expression parameters $\beta_i$. In Sect 3.4 and Fig 2, we show this phenomena of background expression dependence in practice; a radiance field that changes the appearance of points on the background as the view is kept constant but the expression changes. In order to counter this effect, we use a spatial prior on rays that's induced by the fitted FLAME face morphable model. First, we render the FLAME mesh giving us a binary silhouette image $S_i$ for each frame $i$. Next, we define two sets of points, let $R_i^{\text{in}} = \mathbf{x}_1, ..., \mathbf{x}_n$ be the set of points that lie on rays inside the FLAME silhouette of frame $i$ and $R_i^{\text{out}} = \mathbf{x}_1, ..., \mathbf{x}_m$ be the set of points that lie on rays outside the FLAME silhouette of frame $i$ (as shown in Fig 4), the radiance field is now defined as:

$$F : (\boldsymbol{\gamma}(D(\mathbf{x}, \omega_i)), \boldsymbol{\gamma}(\mathbf{d}), \phi_i, \mathbb{I}(\mathbf{x})\beta_i) \to (\mathbf{c}(\mathbf{x}, \mathbf{d}), \sigma(\mathbf{x})) \tag{5}$$

where, $\mathbb{I}(\mathbf{x}) = 1$ if $\mathbf{x} \in R_i^{\text{in}}$ and 0 otherwise . This ensures that points that do not affect face pixels are not affected by facial expression changes. As can be seen in Fig 2, such a spatial ray prior effectively disentangles the appearance and expression and ensures that the background is unaffected by facial expression parameters.

**Face region regularization.** Since our method optimizes extrinsic camera parameters w.r.t the FLAME 3DMM, we assume the head is static and has an identity mapping to the canonical frame. In order to prevent the deformation network, $D$, from moving the 3DMM, we penalize any deformation on points sampled from it as follows:

$$\min_{\psi, \omega_i} ||D(\mathbf{x}_{\text{3DMM}}; \omega_i) - \mathbf{x}_{\text{3DMM}}||; \quad \forall i \tag{6}$$

where, $\psi$ are the parameters of $D$.

### 3.4 On the necessity of a Spatial Ray Prior

In this section we demonstrate the necessity of the FLAME induced spatial prior on ray sampling as discussed in Sect 3.3. In Fig 2, we show the results of renanimating a portrait video with a constant view directions using our method with and without a spatial ray prior. As can be seen in Fig 2, the results of the model without the prior are of lower quality than that of the model with it. Additionally, when we calculate the difference image of the reanimated frames generated by both methods we see that, *despite the viewing direction remaining constant*, the model without the prior changes the background. In stark contrast, and unsurprisingly, the model with the spatial ray prior does not do so and only makes changes around the face. The prior explicitly disentangles the expression parameters and the appearance in regions of the scene which do not project to the face. Therefore, we see that the spatial prior is a necessary ingredient high quality portrait video reanimation.

## 4 Results

In this section, we show results of facial expression control and novel view synthesis on portrait videos captured using a standard smartphone. Since our work is the first to enable both explicit dynamic face controls and novel view synthesis for portrait videos, there is no existing work for an apple-to-apple comparison. We qualitatively and quantitatively compare our method to three baseline approaches that perform similar tasks: (1) Nerfies (Park et al., 2021): a state-of-the-art method using NeRF for novel view synthesis on static portrait scene, (2) NerFACE Gafni et al. (2021), a state-of-the-art method using NeRF for face dynamics control without modeling the entire scene, and (3) First Order Motion Model (FOMM) (Siarohin et al., 2019), a general-purpose image reanimation pipeline. We use the appearance code of the first frame for FLAME-*in*-NeRF, Nerfies (Park et al., 2021) and NerFACE (Gafni et al., 2021) to perform the reanimation. For FLAME-*in*-NeRF and Nerfies (Park et al., 2021), we use the deformation code from the first frame. Full videos of the reanimation can be found in the supplementary material. We strongly urge the readers to check them out to see FLAME-*in*-NeRF performing at its best.

**Training Data Capture and Training details.** The training data was captured using an iPhone XR smartphone for all the experiments in the paper. We ask the subject to enact a wide range of expressions and speech while trying to keep their head still as the camera is panned around them. Alternatively, the subject can self-capture a video (a selfie video) as they speak. We calculate the expression and shape parameters of each frames in the videos using DECA (Feng et al., 2021). Next, we compute the extrinsic camera parameters via standard landmark fitting using landmarks predicted by (Guo et al., 2020). All training videos are about just 25 seconds long ($\sim$ 700 frames). Due to compute restrictions, the video is down-sampled and the models are trained at 256x256 resolution. We use coarse-to-fine regularization (Park et al., 2021) to train the deformation network $D(\mathbf{x}, \omega_i)$. Please find full details of each experiment in the Section 3. of the supplementary.

### 4.1 Evaluation on Validation Data

We evaluate FLAME-*in*-NeRF, Nerfies (Park et al., 2021), NerFACE (Gafni et al., 2021) and FOMM (Siarohin et al., 2019) on held out images. Since FLAME-*in*-NeRF and Nerfies (Park et al., 2021)

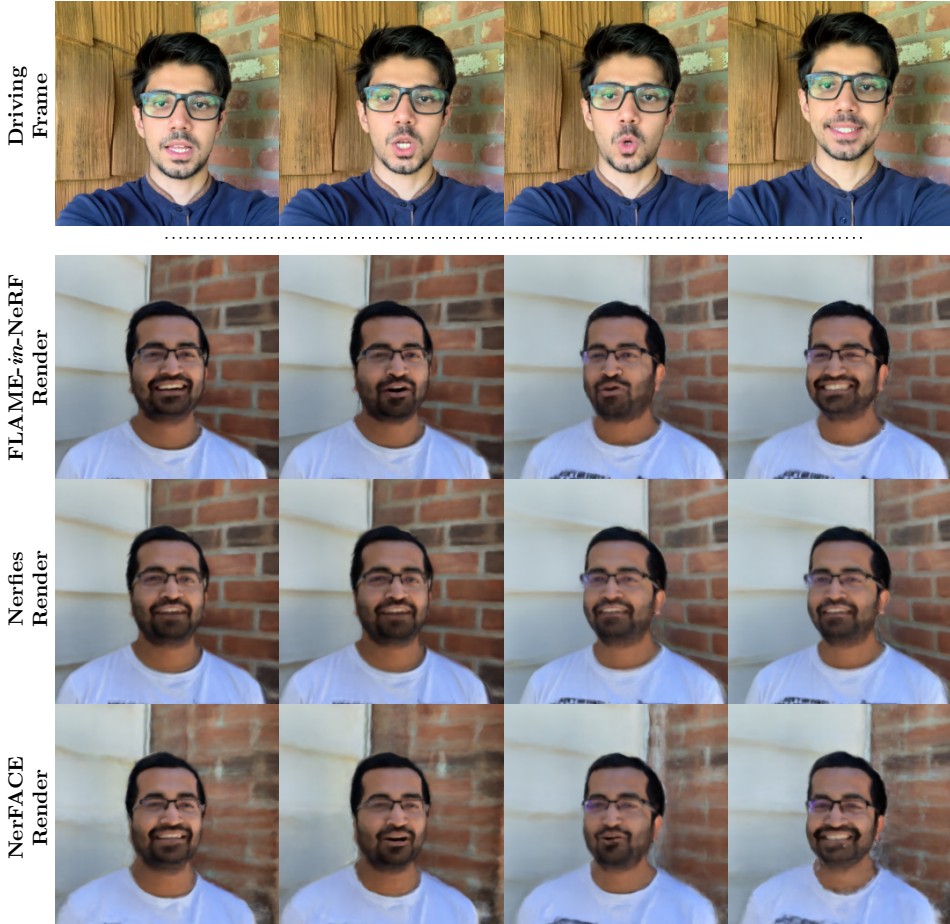

Figure 5: **Qualitative evaluation by reanimating subject 1**: Here we show the results of reanimating Subject 1 using our method, Nerfies (Park et al., 2021) and NerFACE Gafni et al. (2021) with both expression and view changes. The first row shows the driving frame, the second row shows the results of our method, the third shows the results of Nerfies (Park et al., 2021) and the fourth row shows the results of NerFACE (Gafni et al., 2021). As can be seen in columns 1-4, Nerfies (Park et al., 2021) is unable to model the facial expressions correctly leading to artefacts on the face while NerFACE (Gafni et al., 2021) is unable to model the 3D scene leading to artefatcs on the background. We see that our method, in contrast with Nerfies and NerFACE, generates high-quality reanimation results with high fidelity to the driving expression and consistency across views.

use a per-frame deformation and appearance code, $\omega_i$ and $\phi_i$ respectively, we cannot perform a direct comparison with the ground truth image as it may have a different deformation to the canonical frame than the first frame (which is what we use as default for reanimation). Therefore, we first find the deformation of a given validation image to the canonical frame by optimizing the deformation code as follows

$$\min_{\omega} ||C_p(\omega; \mathbf{x}, \mathbf{d}, \theta, \phi_0) - C_p^{GT}|| \tag{7}$$

where, $C_p(\omega; \mathbf{x}, \mathbf{d}, \theta, \phi_0)$ is the predicted color at pixel $p$ generated using Eq. (1) and Eq. (5), $\phi_0$ is the appearance code of the first frame, $\theta$ are the parameters of $F$ as defined in Eq. (5) and $C_p^{GT}$ is the ground-truth pixel value. Note, we *only* optimize $\omega$, all other parameters of the radiance field are kept fixed. We optimize Eq. (7) for 2000 epochs which we observe to be more than enough to find the loss plateau. Once the optimization finishes, we report the final MSE, PSNR, LPIPS and Face MSE i.e the MSE only over the face region. We use no such optimization with NerFACE (Gafni et al., 2021) since it does not have a deformation module and on FOMM (Siarohin et al., 2019) since it is a image-based method. As can be seen in Table 1, our method outperforms Nerfies (Park et al., 2021), NerFACE (Gafni et al., 2021) and FOMM (Siarohin et al., 2019) on validation images when

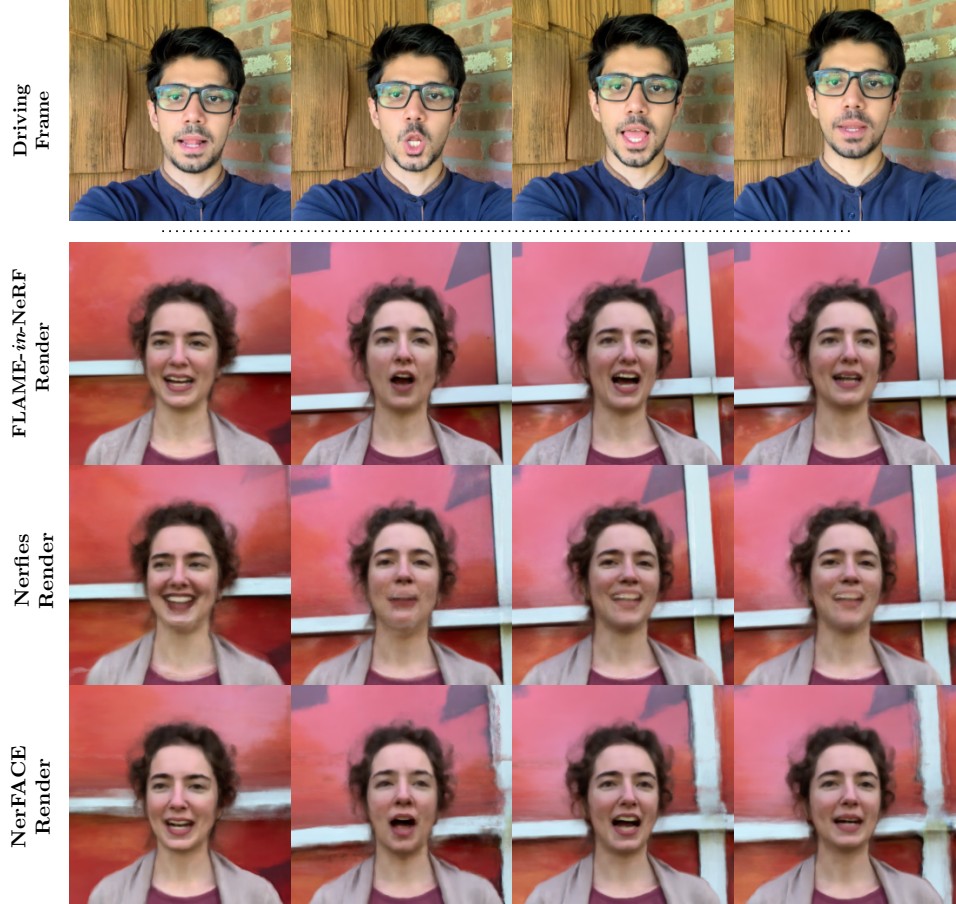

Figure 6: **Qualitative evaluation by reanimating subject 3**: Here we show the results of reanimating Subject 3 using our method, Nerfies (Park et al., 2021) and NerFACE (Gafni et al., 2021) with both expression and view changes. The first row shows the driving frame, the second row shows the results of our method, the third shows the results of Nerfies (Park et al., 2021) and the fourth row shows the results of NerFACE (Gafni et al., 2021). As can be seen in columns 1-4, Nerfies is unable to model the facial expressions correctly leading to artefacts on the face while NerFACE is unable to model the 3D scene leading to artefatcs on the background. We see that FLAME-*in*-NeRF, in contrast with Nerfies and NerFACE, generates high-quality reanimation results with high fidelity to the driving expression and consistency across views.

averaged over all subjects. Since FLAME-in-NeRF, Nerfies and NerFACE are trained on dynamic portrait videos with changing facial expressions, Nerfies is unable to learn the topological changes of the mouth, often regressing to a 'mean' expression (see the third row of Fig 5 and Fig 6) with small view-dependent changes. NerFACE on the hand, lacking a deformation module and a spatial prior, is unable to model the dynamism of the scene and generates significant artefacts in the background 3D scene (see the fourth row of Fig 5 and Fig 6). FOMM (Siarohin et al., 2019), being an image based method, is unable to model novel views. In contrast to Nerfies (Park et al., 2021), NerFACE Gafni et al. (2021) and FOMM (Siarohin et al., 2019), FLAME-*in*-NeRF, due to the use of a spatial prior along with expression conditioning and a deformation module, is able to model both facial expressions and the full 3D scene with high fidelity, thus giving better reconstructions.

## 4.2 REANIMATION WITH ARBITRARY EXPRESSION CONTROL AND NOVEL VIEW SYNTHESIS

In this section we show results of reanimating Neural Radiance Fields using both FLAME-*in*-NeRF and Nerfies (Park et al., 2021) using expression parameters as the driving parameters. Per-frame expression parameters from the driving video are extracted using DECA (Feng et al., 2021) and are given as input to FLAME-*in*-NeRF as follows:

| Subject | Method | MSE (↓) | PSNR (↑) | LPIPS (↓) | Face MSE (↓) |
|---------|--------|---------|----------|-----------|--------------|
| Subject 1 | FLAME-*in*-NeRF | 2.045e-3 | 26.89 | 0.185 | 4.708e-4 |
| | Nerfies (Park et al., 2021) | 2.82e-3 | 25.49 | 0.205 | 8.140e-4 |
| | NerFACE (Gafni et al., 2021) | 8.68e-3 | 20.61 | 0.326 | 1.166e-3 |
| | FOMM (Siarohin et al., 2019) | 2.631e-2 | 15.79 | 0.474 | 4.043e-3 |
| Subject 2 | FLAME-*in*-NeRF | 1.255e-3 | 29.01 | 0.147 | 1.688e-4 |
| | Nerfies (Park et al., 2021) | 2.05e-3 | 26.06 | 0.157 | 2.806e-4 |
| | NerFACE (Gafni et al., 2021) | 4.34e-3 | 23.62 | 0.233 | 3.171e-4 |
| | FOMM (Siarohin et al., 2019) | 3.221e-2 | 14.91 | 0.500 | 6.909e-3 |
| Subject 3 | FLAME-*in*-NeRF | 5.71e-3 | 22.43 | 0.223 | 2.84e-4 |
| | Nerfies (Park et al., 2021) | 4.68e-3 | 23.29 | 0.277 | 7.85e-3 |
| | NerFACE (Gafni et al., 2021) | 6.347e-3 | 21.97 | 0.285 | 3.38e-4 |
| | FOMM (Siarohin et al., 2019) | 5.541e-2 | 12.56 | 0.476 | 7.496e-3 |
| Subject 4 | FLAME-*in*-NeRF | 2.712e-3 | 25.67 | 0.236 | 4.35e-4 |
| | Nerfies (Park et al., 2021) | 5.9e-3 | 22.29 | 0.384 | 1.187e-3 |
| | NerFACE (Gafni et al., 2021) | 4.556e-3 | 23.41 | 0.338 | 5.76e-4 |
| | FOMM (Siarohin et al., 2019) | 3.758e-2 | 14.25 | 0.505 | 8.054e-3 |
| *Average* | FLAME-*in*-NeRF | 2.93e-3 | 25.97 | 0.197 | 3.39e-4 |
| | Nerfies (Park et al., 2021) | 3.862e-3 | 24.28 | 0.255 | 2.532e-3 |
| | NerFACE (Gafni et al., 2021) | 5.98e-3 | 22.40 | 0.295 | 1.359e-3 |
| | FOMM (Siarohin et al., 2019) | 3.787e-2 | 14.37 | 0.488 | 6.625e-3 |

Table 1: Quantitative results of Subject 1,2,3 and 4 on validation data. Our results are better than Nerfies (Park et al., 2021), NerFACE (Gafni et al., 2021) and FOMM (Siarohin et al., 2019) on most metrics across all subjects.

| Method | MSE (↓) | PSNR (↑) | LPIPS (↓) | Face MSE (↓) |
|--------|---------|----------|-----------|--------------|
| FLAME-*in*-NeRF | 2.6e-3 | 25.83 | 0.223 | 3.48e-4 |
| FLAME-*in*-NeRF w/o Spatial Prior | 2.83e-3 | 25.40 | 0.377 | 4.75e-4 |

Table 2: Here we show a quantitative comparison between using the spatial ray prior and not. As can be seen, using the spatial prior gives significantly better results, especially with respect to perceptual quality as measured by LPIPS.

$$F : \left( \boldsymbol{\gamma}(D(\mathbf{x}, \omega_i)), \boldsymbol{\gamma}(\mathbf{d}), \phi_i, \mathbb{I}(\mathbf{x})\beta_i^{\text{Drive}} \right) \to (\mathbf{c}(\mathbf{x}, \mathbf{d}), \sigma(\mathbf{x})) \tag{8}$$

where, $\beta_i^{\text{Drive}}$ are the expression parameters derived from the driving video. We refer the reader to Eq. (5) for the definitions of the other variables. Since Nerfies (Park et al., 2021) do not take as input expression parameters, it's forward pass remains the same as in Eq. (3). As we drive both methods, we also simultaneously change the viewing direction. Fig 5 and Fig 6 shows the results of both FLAME-*in*-NeRF, Nerfies (Park et al., 2021) and NerFACE (Gafni et al., 2021) with changing the expression parameters and view on two different subjects. As one can see, FLAME-*in*-NeRF captures the driving expression with high fidelity and is able to do so regardless of viewing direction and without compromising the reconstruction of the background 3D scene. Nerfies (Park et al., 2021), lacking the explicit conditioning on expression parameters, is unable to generate images with the correct facial expression. Only small changes in expression can be observed due to view changes. NerFACE (Gafni et al., 2021), lacking a deformation model, cannot model a dynamic scene as can be seen by poor reconstruction of the background in Fig 5 and Fig 6.

## 5 CONCLUSION

In this paper we have presented FLAME-*in*-NeRF, a novel method capable of arbitrary facial expression control and novel view synthesis for portrait videos. FLAME-*in*-NeRF uses an expression-conditioned neural radiance field along with a spatial prior to generate images with high fidelity to both the subject in the original portrait video and the provided expression parameters, in any viewing direction. FLAME-*in*-NeRF is also able to model details of the subject's face such as hair and glasses and reproduce them with high fidelity as the video is driven. However, the problem of controllable human head models with novel view synthesis is far from solved. FLAME-*in*-NeRF

is unable to model large head movements and requires the subject in the portrait video to remain relatively still. We hope to address this in future work. These are exciting times to be a part of ML/CV/CG communities as neural methods push the state of the art in head control and novel view synthesis with broader impacts in entertainment, education and HCI.

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
