# OpenReview forum: "FLAME-in-NeRF: Neural control of Radiance Fields for Free View Face Animation"
_ICLR.cc/2022/Conference — ICLR 2022 Submitted_

### Official Review · Reviewer_A3Tv · 2021-10-24

**Correctness:** 3
**Technical Novelty And Significance:** 2
**Empirical Novelty And Significance:** 2
**Recommendation:** 3
**Confidence:** 4

**Details Of Ethics Concerns:**

-

**Main Review:**

Strength:
+ Unlike previous works, the proposed framework enables explicit control of the human facial expression while synthesizing an image from a novel viewpoint.
+ The writing is clear and easy to follow.


Weakness:
- Lack of novelty. The key idea, i.e., combining foreground masks to remove the artifacts from the background, is not new. Separate handling of foreground from background is a common practice for dynamic scene novel view synthesis, and many recent methods do not even require the foreground masks for modeling dynamic scenes (they jointly model the foreground region prediction module, e.g., Tretschk et al. 2021).

- Lack of controllability. 1) The proposed method cannot handle the headpose. While this paper defers this problem to a future work, a previous work (e.g., Gafni et al. ICCV 2021) is already able to control both facial expression and headpose. Why is it not possible to condition the headpose parameters in the NeRF beyond the facial expression similar to [Gafni et al. ICCV 2021]? 2) Even for controlling facial expression, it is highly limited to the mouth region only. From overall qualitative results and demo video, it is not clear the method indeed can handle overall facial expression including eyes, nose, and wrinkle details, and the diversity in mouth shape that the model can deliver is significantly limited.

- Low quality. The results made by the proposed method are of quite low quality. 1)  Low resolution: While many previous works introduce high-quality view synthesis results with high-resolution (512x512 or more), this paper shows low resolution results (256x256) for some reasons. Simply saying the problem of resources is not a convincing argument since many existing works already proved the feasibility of high resolution image synthesis using implicit function. Due to this low-resolution nature, many high-frequency details (e.g., facial wrinkles), which are the key to enabling photorealistic face image synthesis, are washed out. 2)  In many cases, the conditioning facial expressions do not match that from the synthesized image. From the demo video, while mouth opening or closing are somehow synchronized with conditioning videos, there exists a mismatch in the style of the detailed mouth shape.

**Summary Of The Paper:**

This paper introduces a method for novel view synthesis of a dynamic scene with a person in varying facial expression. To enable the controllability of facial expression, a parametric 3D face model is combined with existing neural radiance field-based approaches. To overcome background artifacts (e.g., distortion) while animating, the spatial face prior is added in a form of binary mask. The method is evaluated on four customized videos.


**Summary Of The Review:**

While this paper add a controllability in the facial expression for a novel view synthesis by combining the techniques from previous works, the idea is not novel (which shows many overlaps with many existing NeRF based approaches), and the image synthesis quality is not convincing (low resolution without sufficient high-frequency details, notable jitters, and mismatch between the synthesized facial expression and conditioning ones) when compared to how existing methods have demonstrated. Although the proposed method performs better in several customized scenarios, but those testing examples do not cover diverse scenes that can truly validate the performance. Based on the current demonstration, the ability to control the facial expression is highly limited to a few number of mouth shapes (it's more like mouth control, not facial expression control). Considering the novelty and quality in the context of existing methods, the initial rating is reject.

---

> ### Author Response · Authors · 2021-11-23
> **Response to Reviewer A3Tv**
>
> Thank you for your review. Below we address specific concerns
>
> - **Lack of Novelty:** We would like to note that none of the work you have mentioned adds control of facial expressions to NeRF. We believe this is novel since this is the first work that enables joint modeling of the 3D scene and facial reanimation.
> - **Controllability:** First, we would like to note that NerFACE does not enable novel-view synthesis of the scene, while we do. We discuss this difference in the related works section (see 'Controllable Face Generation'). At this link: [https://anonymous.4open.science/r/FLAME-in-NeRF-ICLR2022-51CD/README.md](https://anonymous.4open.science/r/FLAME-in-NeRF-ICLR2022-51CD/README.md), we show results of eye-control using stills from videos in the supplementary. Further, the mouth shape, as shown in Figs 5 and 6, and in the videos, are individualized and have high fidelity to driving video.
> - **Quality:** Unfortunately, we indeed are constrained by computational resources. Under similar experimental conditions and compute, we have an overall better quality of results than comparable SOTA work (see Table 1).

---

> > ### Comment · Reviewer_A3Tv · 2021-11-29
> > **Reply to authors**
> >
> > Dear authors,
> >
> > thanks for the reply to the comments. After reading the rebuttals, I would like to keep my rating since all my concerns are weakly addressed.
> >
> > For lack of novelty, the responses are not direct to my concern which is lying on the spatial ray prior (not controllability of the facial expression) claimed as a main contribution. For controllability, NerFACE enables pose control by simply conditioning pose parameters in the implicit function. This paper should clearly justify why conditioning pose parameters in the current system is not possible. For quality, the concern is not only about the image resolution but also about others (see the original comments).

---

> > > ### Author Response · Authors · 2021-11-29
> > > **Novelty and Clarification regarding NerFACE**
> > >
> > > Thanks for the reply.
> > >
> > > We would like to note that in the list of contributions (in the introduction) we claim to "develop a system capable of simultaneous control of facial expressions and viewing direction trained on short videos captured from a mobile phone", thus a method for the *joint* modelling of facial expressions (allowing facial expression control) and the *full 3D scene* (allowing novel view synthesis) is indeed a main contribution. Further, we demonstrate the problem of *Expression-Appearance Entanglement* qualitatively in Fig 2 and quantitatively in Table 2 and *propose the spatial ray-prior as a solution to it*. As far as we know, we're the first ones to show the problem exists and propose a solution to it.
> > >
> > > We'd also like to point out that NerFACE *does not* allow pose control by conditioning the implicit function on pose parameters. In Fig 2 of the [paper](https://openaccess.thecvf.com/content/CVPR2021/papers/Gafni_Dynamic_Neural_Radiance_Fields_for_Monocular_4D_Facial_Avatar_Reconstruction_CVPR_2021_paper.pdf), we can clearly see that *only the expression parameters are used as input to the implicit function (please follow the blue arrow)*. This can be seen again in Fig 8 of the [arxiv version of the paper](https://arxiv.org/pdf/2012.03065.pdf). Pose control is obtained by treating a *change in head-pose as a change in camera position*, this can be seen in Fig 2 of the [paper](https://openaccess.thecvf.com/content/CVPR2021/papers/Gafni_Dynamic_Neural_Radiance_Fields_for_Monocular_4D_Facial_Avatar_Reconstruction_CVPR_2021_paper.pdf) (please follow the pink arrows). This can also be confirmed by reading the first paragraph of the Methods section of the paper, where they say:
> > >
> > > *"In addition, we employ the pose parameters (rotation, translation) of the face tracking to transform
> > > the rays into a canonical space that is shared by all frames."*
> > >
> > > This is the reason why NerFACE cannot perform novel view synthesis of the 3D scene, changes in head-pose are modeled as camera extrinsics. In contrast, our work, as far as we know, is the first work that allows both facial expression control *and* novel view synthesis of the full 3D scene. In the related work section of the paper (Section 2), in "Controllable Face Generation", we discuss how FLAME-in-NeRF is different from NerFACE.

---

### Official Review · Reviewer_SS8Y · 2021-11-02

**Correctness:** 3
**Technical Novelty And Significance:** 2
**Empirical Novelty And Significance:** 3
**Recommendation:** 5
**Confidence:** 4

**Main Review:**


**Strenghts:**:
- Writing quality. The paper is well-written and is easy to follow.
- Method. The general idea behind the method - to use a spatial disentanglement mechanism to separate effects of conditioning variables - is reasonable and is definitely worth investigating.
- Experimental evaluation. Authors provide a comprehensive comparison to very recent baselines, and show that both qualitatively and quantitatively their method performs slightly better.

**Weaknesses:**

- Method limitations. Authors do hint at a very important issue which is common for animatable models - which is entanglement among different factors of variation among the control signals.  However, the scope of the solution that they propose seems quite limited: effectively the only separation that one gets is between the background and foreground. This means that there is no mechanism for the model to separate independent expression parameters, and even the disentanglement between the viewpoint and expression is not fully guaranteed. It would be interesting to see animation on sequences where a single expression parameter of the time is varied while others are being fixed, and similarly fixing expression vector while varying the view to see if there are any spurious correlations, in order to better understand the scope of this issue.

- Related work - geometrical prior. The idea of using an explicit geometrical or spatial prior to improve generalization is not particularly novel, even if one only considers the context of neural radiance fields. In particular, most of the work on neural rendering for bodies (NeuralBody, Neural Actor) use SMPL or similar model, which acts as a 3DMM. Although those works do not model the background, ideas presented in those works will likely generalize better as they also encourage spatial disentanglement between different parts of the articulated object. Very similar ideas can be trivially extended to faces data. Moreover, the assumption that the appearance of the background is independent from the expression is not correct - e.g. shadows depend on those?

- Related work - NerFace. Proposed method is very similar to the existing method - NerFACE. The only difference seems to be in the way the two methods treat background? If that is the main motivation for this work, ideally it should be made more clear. It would also be great if authors describe what would be a difference between their method and simply training a separate NERF model for the background (using pre-computed segmentation masks, and simple ad-hoc blending at inference time). I can see that training a joint model might be beneficial, but ideally this should be demonstrated empirically.

- Experimental evaluation. There are no video materials that compare proposed method to the baselines, only static comparisons. I would also encourage authors to provide failure cases of their method to have a better idea about the limitations. As mentioned earlier, varying independent control variables - view, individual expression variables - can provide a way to qualitatively evaluate how animatable the model actually is.

**Misc:**
- P5: able to generate
- P8: artefatcs

**Summary Of The Paper:**

This paper introduces a method for creating animatable portraits within a static scene based on dynamic neural radiance fields.
The main idea is to fit a 3DMM-like model to a set of portrait frames and use it to rasterize a binary mask which is used to enforce distanglement between the foreground and background when training a dynamic NERF. Both qualitative and quantitative results seem to be convincing.

**Summary Of The Review:**

Although qualitative results of the paper seem convincing, the method itself is very similar to an existing one (NerFACE), while the main claimed novelty - the spatial ray prior - seems to be equivalent to training a separate background model with pre-computed segmentation masks. Thus I am leaning more towards rejecting this paper, but I am eager to reconsider my rating if authors address the concerns raised above.

---

> ### Author Response · Authors · 2021-11-23
> **Response to Reviewer SS8Y**
>
> Thank you so much for your thoughtful review! Below we address specific concerns:
>
> - **View and Expression disentanglement:** Here ([https://anonymous.4open.science/r/FLAME-in-NeRF-ICLR2022-51CD/README.md](https://anonymous.4open.science/r/FLAME-in-NeRF-ICLR2022-51CD/README.md)) we show stills from videos *Reanimation_View_Subject_2.avi*, *Reanimation_View_Subject_4.avi* where the subjects are reanimation while being viewed from two different views. As one can see, the expressions reanimated are consistent across views. Thus, FLAME-in-NeRF successfully disentangles expressions and views.
> - **Related Work- Geometrical Prior:** Thank you so much for these references. As you mentioned, those works do not model the full 3D scene and thus have a more limited scope than our paper. However, we do agree, utilizing more geometric priors is fruitful avenue for further work and we believe FLAME-in-NeRF is a first step towards using geometric priors for fully re-animatable 3D portrait scenes.
> - **Training two NeRFs:** It does make intuitive sense to train two NeRFs, however, we believe our approach is likely to work better since it explicitly induces a semantic separation of objects within the 3D by using the spatial prior and does not need ad-hoc blending which can be prone to artefacts.
> - **Videos of other methods:** We will include more videos in the supplementary of the paper upon acceptance.

---

> > ### Comment · Reviewer_SS8Y · 2021-11-24
> > **Re:**
> >
> > Thanks for the response. I am still not fully convinced by authors' arguments with respect to related work. I believe that those methods are very relevant and do provide a somewhat more comprehensive solution to using geometrical priors, compared to the one presented in this work. Thus I am currently inclined to keep my original rating.

---

> > > ### Author Response · Authors · 2021-11-30
> > > **Comparison to an untested Hypothesis**
> > >
> > > The reviewer speculates that the prior from NeuralBody/Neural Actor might work in the problem for which our method provably works. However there is no evidence that this is the case neither in the cited papers nor in the reviewer's assertions. We urge the reviewer and the AC to judge the paper based on the published record instead.

---

> > > > ### Comment · Reviewer_SS8Y · 2021-11-30
> > > > **Re:**
> > > >
> > > > Thank you for the reply! Sorry if I was not clear, but I would like to disagree with the "there is no evidence" statement. I do believe that the aforementioned prior works absolutely do provide evidence for the efficiency of geometrical priors (admittedly in a slightly different context). At the same time, the method presented in this paper provides much weaker evidence for the efficiency of geometrical prior in general, and rather demonstrates that adding background modeling to NerFACE improves its performance on scenes with largely static background. It might certainly be a valid contribution, and a very useful trick in practice, but I am not sure that it is: a) non-trivial and novel, b) adequately stated in the paper as such, c) proven necessary (no comparison with a trivial "two nerf + segmentation mask" baseline). Please also note that similar concerns have been raised by other reviewers.

---

> > > > > ### Author Response · Authors · 2021-11-30
> > > > >
> > > > > Thanks for the clarification! Unlike our work, *neither* of those works jointly models the background and the dynamic object *simultaneously*. Thus, there is no evidence that using such a geometric prior in a joint modelling setting will work. In the paper, we demonstrate the problem of Expression-Appearance entanglement which arises from concatenating the driving parameters to the dynamic NeRF. Additionally, we also propose a solution to it by using a spatial ray-prior.  We believe this is novel contribution to the community and, as far as we are aware, there is no other work that enables control over facial expressions *and* novel view synthesis of the 3D scene.
> > > > >
> > > > > Vis-a-vis a "trivial" Two-Nerf model, we would like to note that such model is again, hypothetically, a solution to our problem. It is not a "trivial" baseline. We chose to use a spatial-ray prior because it does not require two NeRF models and does not need to additionally predict the segmentation mask when the scene is viewed from a novel view. Prediction of the segmentation mask for novel views (which can be prone to artefacts) is *non-trivial* and necessary to accurately blend the *dynamic* foreground and background because humans are never truly static (also demonstrated in Nerfies Park et al). The Two-NeRF model is an alternative, and for now a hypothetical, method to solve our problem. As said earlier, we still urge the reviewer and the AC to judge the paper based on the published record.

---

### Official Review · Reviewer_G8XN · 2021-11-03

**Correctness:** 3
**Technical Novelty And Significance:** 2
**Empirical Novelty And Significance:** Not applicable
**Recommendation:** 3
**Confidence:** 4

**Main Review:**

Strengths:
1. The submitted paper demonstrates the interesting use case of controlling video synthesis with a driving frame/video as input.

Weaknesses:
1. Missing important baseline/citation: HyperNeRF is an improvement over Nerfies and should be considered as a baseline method: "HyperNeRF: A Higher-Dimensional Representation for Topologically Varying Neural Radiance Fields"
2. Quality of synthesized videos: the synthesized videos are low-resolution, blurry, and there are noticeable floaters around human subjects. I don't consider the quality to be on par with Nerfies or HyperNeRF.
3. Limited novelty: the proposed method is a relatively straightforward extension of Nerfies but with masks and facial expressions as the latent code.
4. Limited evaluation: there are only 4 subjects being used as evaluation data

**Summary Of The Paper:**

The submitted paper focuses on novel view synthesis from portrait video with controllable facial expressions. The authors separate the modeling of the static background from the dynamic foreground by segmenting out the faces (using landmark detection and fitting). The faces are modeled with Nerfies with expression parameters as latent codes, while the static background is essentially modeled with a standard NeRF without conditioning on per-frame expressions. The proposed method was evaluated on 4 subjects with data collected from a phone. The authors also demonstrate an application of the proposed method where videos can be synthesized with consistent facial expressions as a driving video as input.

**Summary Of The Review:**

My current recommendation is "3: reject, not good enough". There is limited novelty with important baseline missing (HyperNeRF). The synthesized videos are also blurry with artifacts.

---

### Official Review · Reviewer_z2xe · 2021-11-04

**Correctness:** 3
**Technical Novelty And Significance:** 2
**Empirical Novelty And Significance:** 2
**Recommendation:** 5
**Confidence:** 5

**Main Review:**


This paper presents a straightforward but sensible method that benefits the best of both worlds: it utilizes the very strong prior of human faces by using a parametric face model and meanwhile uses the powerful NeRF to handle complex appearance, including that of the background. The quantitative experiments suggest that the proposed method outperforms the baseline methods. Qualitatively, it achieves something that prior methods don't: It allows for expression control that Nerfies don't support, and it handles the background better than NerFACE because it disentangles expressions from backgrounds.

Now the weaknesses:

Since the method relies heavily on how well DECA and landmark fitting work, it should be discussed whether there is any measurement taken to recover from DECA's and/or landmark fitting's mistakes. If I understand correctly, there is no such error recovery scheme in the model design. This seems like a major flaw to me. Just to name one potential failure that DECA may suffer from: The camera extrinsics from DECA may not be accurate, and most of us would agree NeRF-like models rely on good camera poses to yield sharp rendering.

The video results look rather blurry to me maybe partially because the model was trained at 256x256. I think for the method to be really useful, the authors should explore why such blurriness still exists with the ray sampling masking scheme and the deformation fields already in place. There are also flickering artifacts that "morph" the person's head when the expressions are driven.

I believe there is complex dynamics among the expression parameter beta_i, the deformation code omega_i, and the appearance code phi_i: For instance, how does beta_i interact with omega_i? Intuitively, when the expression changes, the deformation field will have to adjust itself to work with the new expression. This dynamics, or rather this entanglement, doesn't sound desirable to me. This deserves at least some discussions, if not experiments.

Equation 6: Why wouldn't the model get into the trivial solution of having a no-op deformation field D? That would give you a perfect zero loss for Equation 6.

It's strange to have 3.4 ON THE NECESSITY OF A SPATIAL RAY PRIOR as a separate section than 3.3.

There doesn't seem to be any quantitative ablation study. Although we can see qualitatively the disentanglement, it'd be nice to confirm this with numbers.

Fig. 5 shows really similar-quality results between the proposed method and NerFACE in terms of just the reenactment, although I agree the proposed method does perform better overall with fewer artifacts. That said, this observation, together with the fact that the reenacted expressions don't resemble those in the driving frames (e.g., the expressions in Driving Frames 1 & 4 are very different, but they both lead to similar reenacted expressions), casts it into question whether the proposed method controls the expressions well.

Missing references:
- In the human domain, the authors should cite:
    Neural Light Transport for Relighting and View Synthesis
  which does perform view synthesis of human subjects using a "neural rendering" approach.
- Furthermore, this paper:
    Deep Relightable Textures Volumetric Performance Capture with Neural Rendering
  should also be cited, since it does handle moving subjects.
- I think there should be a section on "Parametric Face Models" that are dedicated to 3DMM, FLAME, etc. since half of the model name "Flame-in-NeRF" lies in that domain. The current Related Work sections focus on just the "NeRF half."

Minor nits:
- Page 2: "Lassner & ZollhÃ¶fer"
- Page 3, grammatical errors: "Nerfies (Park et al., 2021) too work", and then "frame, however it"
- Page 5: "are able generate" & "this phenomena of"
- Fig. 4 seems unnecessarily verbose since Eq. 5 already says the same thing very clearly.

**Summary Of The Paper:**


This paper proposes a method that marries parametric face models and neural radiance fields to achieve view synthesis of a portrait that provides expression control. Another view of this work is extending Park et al.'s Nerfies to support expression control and driving.

The paper makes three contributions: Firstly, it proposes a dynamic NeRF model that supports certain explicit control. Secondly, the proposed method disentangles the facial expression from the appearance of the background with a simple masking scheme dubbed as spatial ray sampling prior. Finally, the method enables the simultaneous control of viewpoints and facial expressions.

**Summary Of The Review:**


This is a paper that clearly explains a straightforward but sensible approach. I judge it to be slightly below the acceptance bar mainly because (1) the method heavily depends on the success of two non-trivial preprocessing steps and has no way to recover from their mistakes, and (2) the complex dynamics among the expression parameters and deformation fields remains unexplored.

---

> ### Author Response · Authors · 2021-11-23
> **Response to Reviewer z2xe**
>
> Thank you so much for taking the time to review our paper! Below we address specific concerns:
>
> - **Reliance on DECA for camera calibration:** Like you said, NeRF-like methods rely on good camera calibration for sharp results, which is why we choose to use DECA instead of COLMAP to calibrate our cameras. COLMAP fails on on videos of Subject 3 and Subject 4 (due specularities in the background; a well known weakness of COLMAP, see here: [https://colmap.github.io/tutorial.html](https://colmap.github.io/tutorial.html)). We find that DECA is quite robust and is able to give us good camera calibration. Recovery from bad camera-calibration in NeRF-like settings is an area of active research in itself (see BARF: [https://chenhsuanlin.bitbucket.io/bundle-adjusting-NeRF/](https://chenhsuanlin.bitbucket.io/bundle-adjusting-NeRF/), ICCV2021) and is orthogonal to contributions of the paper.
> - **Blurriness of videos:** We believe that sampling more points along the rays would give us better results. We sampled only 64 points due to computational constraints. We would like to note that other methods generate worse visual quality, as measured by LPIPS, on average than FLAME-in-NeRF when restricted to the same settings (see Table 1).
> - **Entanglement of $\beta_{i}$ and $\omega_{i}$ and Eq 6:** Since we calibrate our cameras with respect to the face, the face itself is assumed to be static while the background is dynamic. Thus, the deformation field (which models dynamism) must be 0 on the face (i.e on the 3DMM), Eq 6 enforces this constraint. Eq 6 *does* not force the global deformation to be zero. The deformation is free to take any value that is necessary to  ensure photometric consistency on the background. Since the deformation value is zero on the face and expressions are modeled through F (see Eq 5), they are disentangled.  We will add this discussion in the paper.
> - **Ablation Study on disentanglement:** Thank you for the suggestion! We have included the results in Table 2 of the paper (highlighted in green on Page 9).
> - **Control of Expression:** The articulation of expression for every person is different. Thus, it is not reasonable to expect EXACTLY the same re-en for all subject. In fact, FLAME-in-NeRF is able to capture this per-person specificity of expression as it re-animates them. We show more results of eye control using FLAME-in-NeRF here: [https://anonymous.4open.science/r/FLAME-in-NeRF-ICLR2022-51CD/README.md](https://anonymous.4open.science/r/FLAME-in-NeRF-ICLR2022-51CD/README.md)
> - **Missing References:** Thank you so much for providing there. We will add them to the paper.

---

### Decision · Program_Chairs · 2022-01-20

**Decision:**

Reject

**Comment:**

This submission received 4 ratings, all below the acceptance threshold. The reviewers expressed concerns around overall novelty of contributions and quality of produced results, and also pointed out lack of comparisons with some prior works and gaps in empirical evaluation. The authors responded to most of these comments, but did not convince the reviewers to upgrade their ratings.
The final recommendation is therefore to reject.